# Influence of CDFW Process Parameters on Microstructure and Mechanical Properties of U75V Rail Steel Welded Joint

**Han Zhang** [1,2] , **Chang'an Li** [3] **and Zhiming Zhu** [1,2,*]

1   Department of Mechanical Engineering, Tsinghua University, Beijing 100084, China;
    zhangh19@mails.tsinghua.edu.cn
2   Key Laboratory for Advanced Materials Processing Technology, Ministry of Education,
    Department of Mechanical Engineering, Tsinghua University, Beijing 100084, China
3   Institute of Materials Joining, School of Materials Science and Engineering, Shandong University,
    Jinan 250061, China; lcasdu@hotmail.com
*   Correspondence: zzmdme@tsinghua.edu.cn; Tel.: +86-139-1079-5346

**Abstract:** In the present paper, the continuous-drive friction welding (CDFW) technology has been successfully applied to join the U75V rail steel. The base metal (BM) of U75V rail steel is lamellar pearlite, and the weld zone could be clearly divided into three subzones (i.e., heat affected zone, thermo-mechanical affected zone (TMAZ), and central weld zone (CWZ)). Electron back-scattered diffraction examinations revealed the martensitic evolution in TMAZ and CWZ, suggesting that the experienced high temperature, severe plastic deformation, and fast cooling rate induce the microstructure transition during the CDFW process. The hard and brittle martensite structure explains the raised microhardness profiles and the reduced impact absorption energy of the as-welded joints. The CDFW process parameters govern the joint properties via influencing the welding heat input and plastic deformation by spindle speed and friction pressure at the friction stage, and the plastic deformation layer (flash) extrusion by upsetting pressure at the upsetting stage. More favorable results could be obtained at small set values of spindle speed (1800 rpm) and friction pressure (75 MPa) with less heat input and plastic deformation, and a large set value of upsetting pressure (175 MPa) with more flash extrusion, whose tensile strength reached 94.3% of that of the BM.

**Keywords:** continuous-drive friction welding; rail steel; microstructure; tensile strength; microhardness; impact toughness





## 1. Introduction

With the ever-increasing demand for passenger and cargo transportation in China, the railway system is evolving toward high-speed and heavy-haul, respectively. This puts forward increasing requirements for the mechanical properties of the steel rail as well as its welded joint and the size accuracy of the integral seamless railway line, as they are the foundation for the development of a high-level railway system [1,2]. The seamless rail lines usually join from long fixed-length steel rails by some proven and applicable welding methods. It is precisely because of the widespread application of long fixed-length steel rails with excellent mechanical properties and some effective welding methods with ensured welded joint performance that the safety and stability of high-speed and heavy-haul trains are effectively guaranteed, the service life of the wheel–rail system is more prolonged, and the passenger comfort is further improved [3,4].

The joining of long fixed-length steel rails generally adopts welding methods such as stationary flash butt welding (FBW) [5] in a factory, while mobile FBW, gas pressure welding, and aluminothermic welding occur at the rail track laying site [6,7]. Additionally, some researches and applications of active gas metal arc welding [8] and narrow gap arc welding [9] have also been reported. However, these welding methods are prone to producing some degree of difference in the mechanical properties between the rail

welded joint and the high-strength steel rail itself. For example, a wide weld zone (tens of millimeters) with both hardening and softening subzones usually exists in the FBW rail welded joints [10]. They also may cause weld defects such as flat spot, white spot, slag inclusions, overburn, and other metallurgical problems [11]. Numerous surveys have demonstrated that the rail welded joints are major vulnerable areas in continuously welded rail (CWR) lines [12]. The vast majority of the incidents (e.g., abrasion, low collapse, crack, and even fracture) come from the weld zones, which severely impacts the durability and service life of the entire CWR lines [13]. Moreover, to extend the service life and the unit load of the CWR lines, the materials of steel rail have also developed from pearlite steel into the steels with higher strength, higher toughness, and better wear resistance (e.g., ultra-fine pitch pearlite, bainite, and bainite/martensite duplex-phase steels) [14,15]. All of these above present more significant challenges to the welding of steel rail.

The solid-state welding technique with less welding heat input is able to suppress the width of the welded joint zone and the eutectoid transformation, thus making a sound welded joint with excellent mechanical properties. Continuous-drive friction welding (CDFW) is a basic mode of a solid-state welding method with the advantages of high joining quality, high production efficiency, energy-saving, and pollution-free [16,17]. It has been widely used in the realms of aviation, aerospace, oil drilling, automobile manufacturing, etc. [18,19]. CDFW has successfully welded numerous kinds of materials (e.g., various steels [20,21], aluminum alloys [22], titanium alloys [23], nickel-based superalloys [24], etc.). These studies concentrate on the influence of CDFW process parameters on the microstructure and mechanical properties of the as-welded joints. For example, Li et al. [25] studied the effect of spindle speed and axial pressure on the formation of CDFW joint of a kind of medium carbon steel. The results show that the welding area temperature as well as the axial shortening length increase with the increase in the spindle speed, and the axial shortening length is positively correlated with the axial pressure. Still, the maximum temperature of the joining interface decreases when the axial pressure is boosted.

The CDFW plays a vital role in the scenes where fusion welding methods may bring some metallurgy-associated problems or are challenging to apply for some materials. For example, the super duplex stainless steel UNS S32760 benefits from high strength and strong stress corrosion cracking resistance as the balanced and homogeneous phase distribution of ferrite and austenite, while the fusion welding and solidification processes are prone to destroying the duplex microstructure. Udayakumar et al. [26] tried to apply CDFW to join UNS S32760 steel and investigated the effect of process parameters such as spindle speed, friction pressure, and upsetting pressure on the welded joint properties, and sound welded joints without secondary phase were eventually obtained. In order to reduce the incompatibility problems (i.e., the generation of intermetallic compounds and residual stress in weld zone) of dissimilar materials, Khidhir et al. [27] applied CDFW in the joining of AISI 1045 medium carbon steel/AISI 316L austenitic stainless steel and obtained 90% joint efficiency. They also discussed the effect of upsetting pressure on the properties of CDFW joints. In these studies, characteristics such as tensile strength, impact toughness, microhardness, and microstructural aspects exhibited by CDFW joints were compared with the parent materials for the evaluation of the welded joint performance and optimization of the process parameters. Besides, there exist CDFW applications on large-size workpieces joining (e.g., Lei et al. [28]) of CDFWed D50Re steel with a diameter of 100 mm.

Due to the remarkable advantages of solid-state welding techniques such as CDFW, it is highly recommended for future steel rail welding. Furthermore, some feasibility studies or conceptual designs have been carried out on applying friction welding technology to the joining of steel rails. Via another variant of the friction welding method, termed linear friction welding (LFW), Su et al. [29] tried to weld cuboid workpieces (with a size of 12 mm × 22 mm × 70 mm, and the section of 12 mm × 22 mm was the interface to be joined) processed from U71Mn pearlite steel rail. Some welding experiments with different process parameters (i.e., friction pressure, swing amplitude as well as frequency, and axial shortening) were conducted. The influence of the process parameters on the welded joints'

performance was analyzed via mechanical property tests and microstructure observations. The maximum tensile strength of the LFW joint with optimal process parameters reached 78.5% of the tensile strength of U71Mn rail steel. Zheng et al. [30] tried to weld the U20Mn bainite rail steel workpieces by LFW. The welding interface of the workpiece was 174.5 mm high (close to the 60 kg/m type rail with a height of 176 mm), and the width was 16.5~30 mm. They obtained LFWed joints without welding defects, and the width of the weld zone was only 4.88~5.69 mm. Furthermore, Johann [31] and Tan et al. [32] put forward patents related to friction welding for steel rail. Their core concept was to insert a middleware between the ends of two steel rails to be joined, moving methods such as rotatory, linear, or orbital modes were used to drive the middleware to generate motion relative to the two steel rail ends, while the two steel rail ends were pressed toward one another along the steel rail longitudinal direction against the middleware. Thus, the welding heat could be generated via the relative friction movement among the middleware and the two steel rail ends. However, these patent methods are only conceptual designs, and their feasibility has not been further revealed. Furthermore, the existing research on friction welding rarely involves the special steel materials of rail steel. Thus, their weldability and welded joint performance by friction welding are worth studying initially.

As the basic mode of friction welding variants (e.g., orbital and rotatory friction welding), CDFW is one of the most mature and widely used methods [18,19]. The ultimate goal of our research work is to develop a new and innovative kind of friction welding method and applicable equipment for steel rail welding to obtain a better welded joint performance. This paper presents the preliminary exploration for applying the CDFW technology to the joining of U75V rail steel and further investigates the influence of CDFW process parameters on the microstructure and mechanical properties of the welded joint. Within this paper, nine groups of experiments with different CDFW process parameters were conducted. The effect of process parameters on the properties of the welded joints was studied and analyzed, and an optimum group of process parameters was considered.

## 2. Material and Experimental Methods

The material to be welded by CDFW is a kind of high-carbon rail steel (pearlite steel), termed U75V. According to the railway industry standard of the People's Republic of China TB/T 2344.1-2020, the chemical composition (wt.%) and mechanical properties of U75V are shown in Table 1. The welding workpieces are cylindrical rods with a diameter of 12 mm and a length of 60 mm. All were processed from the same section of the U75V steel rail in the same direction.

**Table 1.** Chemical composition and mechanical properties of U75V rail steel.

| Chemical Composition (wt.%) | | | | | | Mechanical Properties | | |
|---|---|---|---|---|---|---|---|---|
| C | Mn | Si | V | P | S | Tensile Strength | Elongation after Fracture | Hardness |
| 0.71~0.80 | 0.95~1.05 | 0.50~0.80 | 0.04~0.12 | ≤0.025 | ≤0.025 | $R_m \geq 980$ MPa | $A \geq 10\%$ | 280~320 HB |

The CDFW machine used in this study was HMSZ-4 (Harbin Welding Institute Co. Ltd., Harbin, China) with the maximum spindle speed of 2800 rpm and the maximum upsetting force of 4 t. It is able to weld various cylindrical rods and pipes with a maximum diameter of 30 mm. Generally, the spindle speed (Ss), friction pressure (Fp), time (Ft), upsetting pressure (Up), and time (Ut) are the main process parameters to be set.

Considering the representativeness and our previous studies, for the CDFW experiments of U75V cylindrical rods with a diameter of 12 mm, the setting values of Ss, Fp, and Up were arranged as nine process parameters groups (see Table 2). The set values of Ft and Ut were maintained constant at 3 s and 1 s, respectively.

**Table 2.** Process parameters of CDFW.

| Group | Spindle Speed | Friction Pressure | Upsetting Pressure |
|---|---|---|---|
| 1 | | 75 MPa | 175 MPa |
| 2 | 1800 rpm | 100 MPa | 125 MPa |
| 3 | | 125 MPa | 150 MPa |
| 4 | | 75 MPa | 125 MPa |
| 5 | 2000 rpm | 100 MPa | 150 MPa |
| 6 | | 125 MPa | 175 MPa |
| 7 | | 75 MPa | 150 MPa |
| 8 | 2200 rpm | 100 MPa | 175 MPa |
| 9 | | 125 MPa | 125 MPa |

During the CDFW process, the cylindrical rod (rotating workpiece), which was fixed through the jig equipped on the spindle of the main rotation motor beforehand, started to rotate first. Ss increased to its set value rapidly, then was maintained at a constant. Next, another cylindrical rod (moving workpiece), fixed on the mobile jig in advance, moved slowly toward the rotating rod until the two end surfaces came into contact and applied Fp to its set value. Under the combined action of Ss and Fp, frictional heat was generated between the two contacted end surfaces of the rods. After a lasting period of 3 s (the set value of Ft), termed friction stage, the motor was switched off, Up was axially applied to the two rods, and maintained at its set value with a period of 1 s (the set value of Ut), termed the upsetting stage. Eventually, the CDFW weldments were obtained, as presented in Figure 1.

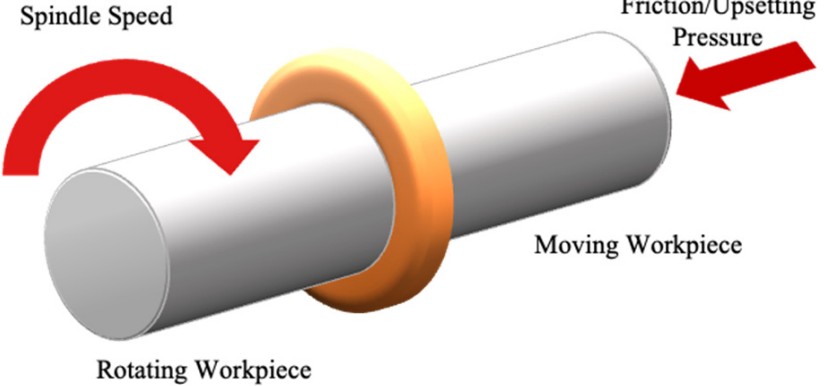

**Figure 1.** Schematic diagram of the CDFW process.

For the hardness test and microstructure observation, the weldments were cut perpendicularly to the weld joining interface. Standard techniques of mechanical grinding and polishing were applied to prepare specimens. Vickers hardness tests were carried out on a FM-800 microhardness tester (Future-Tech Corp., Kawasaki, Japan) under a 500 gF load and a dwell time of 15 s. Fifteen hardness test points with an interval of 0.2 mm along the direction perpendicular to the weld joining interface at the center area of the weldments were achieved in each specimen. For metallographic observation, the specimens were re-polished and corroded with 4% (vt.%) nitric acid alcohol solution for 25 s, and an OLYMPUS BX51RF optical microscope (Olympus Corp., Tokyo, Japan) was used. The microstructure was further examined by a fully automatic electron back-scattered diffraction (EBSD) system Bruker XFlash 6160 (Bruker Corp., Billerica, MA, USA) attached to a Zeiss Gemini SEM 500 scanning electron microscope (Carl Zeiss Corp., Oberkochen, Germany) after precise polishing.

For the tensile test, the weldments were machined according to the Chinese standard GB/T 228.1-2010 and the railway industry standard of the People's Republic of China

TB/T 1632.2-2014 (TB), as depicted in Figure 2a. An Instron 5980 universal testing system (Instron Corp., Norwood, MA, USA) was used to conduct tensile tests with a 2 mm/min tensile rate. The tensile test results could be characterized by tensile strength ($R_m$) and percentage elongation after fracture ($A$). For the Charpy impact test, the weldments were machined according to the Chinese standard GB/T 229-2007 and TB, as shown in Figure 2b, and ZBC series impact testing machine (NSS Laboratory Equipment Co. Ltd., Shenzhen, China) was used. Since the diameter of the welded rod has only 12 mm, it could not meet the size requirement of the standard impact specimen (55 mm × 10 mm × 10 mm), so a small size (55 mm × 10 mm × 5 mm) impact specimen was selected. Then, a pair of 2.5 mm thick gaskets were placed on the specimen holder of the machine to pad up the small size impact specimen, thus making the height center of the small size impact specimen face the pendulum accurately. The results of the Charpy impact test could be characterized by impact absorption energy $KU_{2(5.0)}$, which represents the absorbed energy of the small size impact specimen with a height of 5.0 mm and its U-shape notch with a radius of 2 mm. For fracture analysis, a Leo 1450 Vp scanning electron microscope (Carl Zeiss Corp., Oberkochen, Germany) was used to observe the fracture morphology after tensile and impact tests.

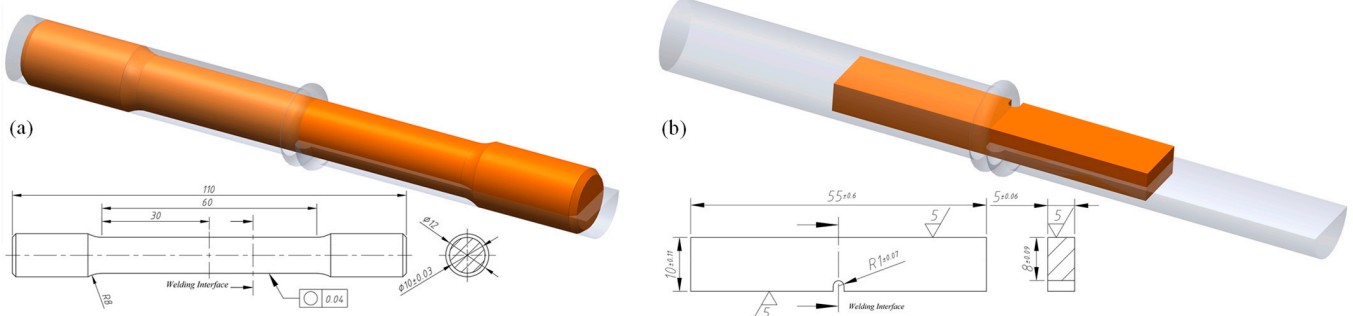

**Figure 2.** Tensile (**a**) and impact (**b**) test specimens.

## 3. Results

### 3.1. Macro-Profile Inspection

The overall photograph of typical weldments corresponding to nine groups of experiments with different process parameters is presented in Figure 3a, where significant curled flashes around the welded joint could be observed, and few defects such as cracks or blowholes were detected for all weldments. The extruded flash is beneficial for reducing weld defects, as it makes the joined interface enclosed, thus effectively avoiding oxidation of the thermoplastic metal during welding and cooling periods. Additionally, the attached grime, contamination particles, and rust on the end surfaces of the cylindrical rods to be joined are entirely removed due to the rubbing action and taken away by extruded flash, termed the "self-cleaning" phenomenon, which further avoids defects such as flat spot and slag, often detected in the FBW joints [18].

The axial shortening amount of the cylindrical rods after the CDFW process is known as burn-off length (BOL). It manifests the degree of softening and plastic deformaion of the joined interfaces and their adjacent areas of the cylindrical rods. Hence, solid-state welding has taken place and is completed [33]. The BOL values of all weldments were measured and given (see Figure 3b). The BOL values varied from 4.28 ± 0.24 mm to 6.05 ± 0.18 mm, and they were much smaller than those of the FBW weldments (usually exceeding 40 mm) [5].

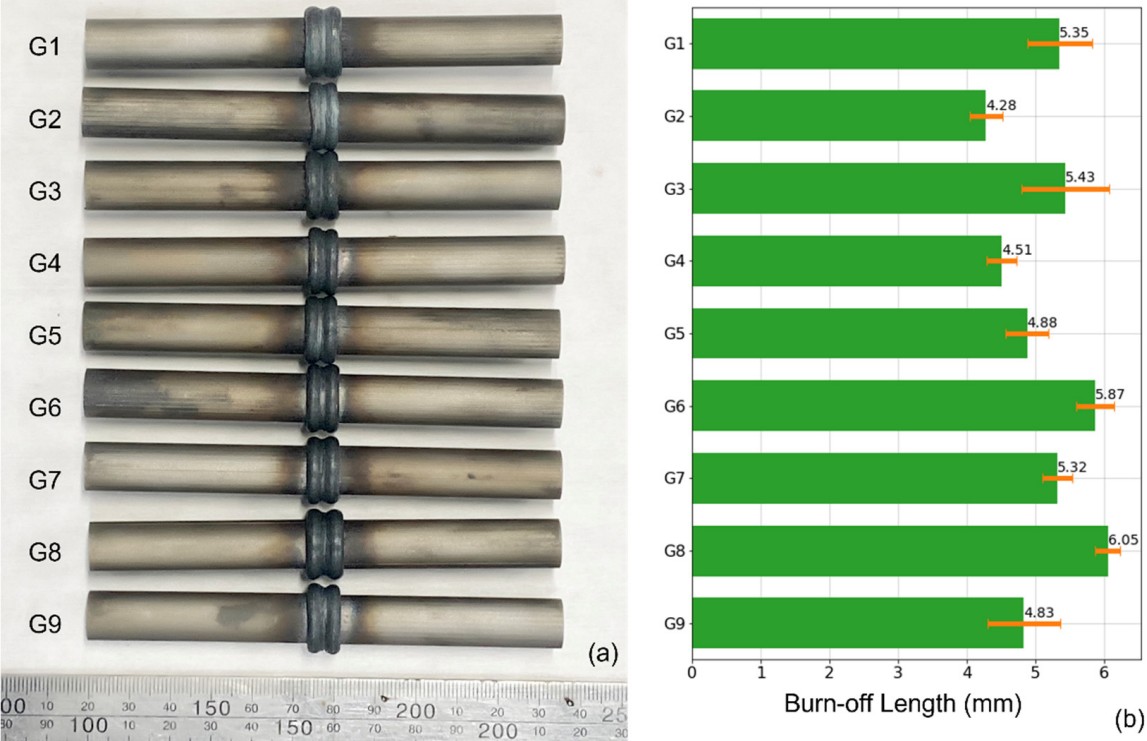

**Figure 3.** Weldments corresponding to nine groups of experiments with different process parameters (**a**), and their BOL values (**b**), from top to bottom, for Groups 1–9, respectively.

### 3.2. Microstructure Observation

The typical microstructures of nine specimens at their welded joint center region are shown in Figure 4. All of these welded joints were well metallurgically bonded without defects such as voids or slags. Compared to the base metal (BM), the microstructure of the weld zone generated significant change, and the relative rotated movement during the CDFW process caused an asymmetric microstructure. The boundary lines of the weld zones at the welded joint center region in Groups 1–3 with a smaller set value of Ss (1800 rpm) are the hyperbolic shape, while the boundary lines of the weld zones in Group 4–9 with higher set values of Ss (2000 or 2200 rpm) are of the linear shape.

The values of weld zone width (WZW) of all specimens were measured during metallographic observation, as shown in Figure 5. The WZW values changed from $1.01 \pm 0.19$ mm to $2.90 \pm 0.11$ mm, and the minimum value of WZW reached only 2.5% of the FBW joint (approximately 40 mm). The variation in the WZW reflects the influence of process parameters on the temperature distribution within the joined interface regions. The WZW value was positively correlated with the set value of Ss. The WZWs of Groups 1–3 were significantly narrower than those of Groups 4–9 as more frictional heat was generated in Groups 4–9 at the being joined interfaces, and higher heat input inclines to widen the weld zone while the WZW was negatively correlated with the set value of Up. Groups 1, 6, and 8, with the largest set value of Up (175 MPa), obtained the narrowest WZW within the 3-Group with the same set values of Ss. This is because the large value of Up is beneficial for the extrusion of thermoplastic metal to form flash, thus thinning the weld zone. It is evident that the flash sizes of Groups 1, 6, and 8 were larger than those of other weldments with the same set values of Ss, as depicted in Figure 3a. The value of Fp had the opposite effects on the WZW.

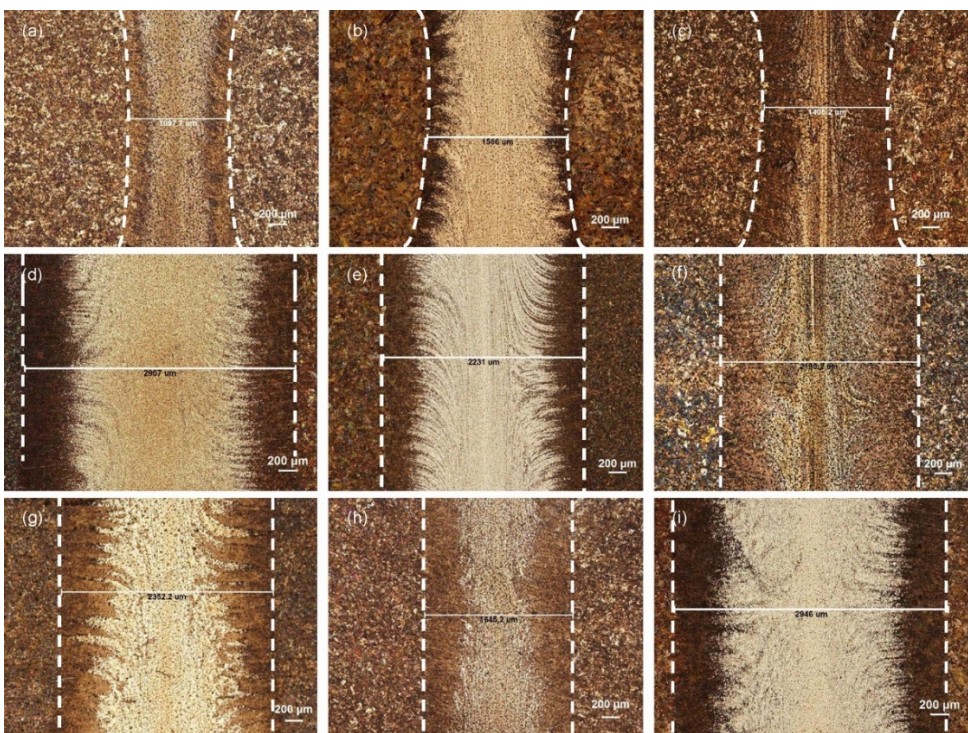

**Figure 4.** Optical micrographs of weld zone for nine specimens in all experimental groups: (**a–i**) represent Groups 1–9, respectively.

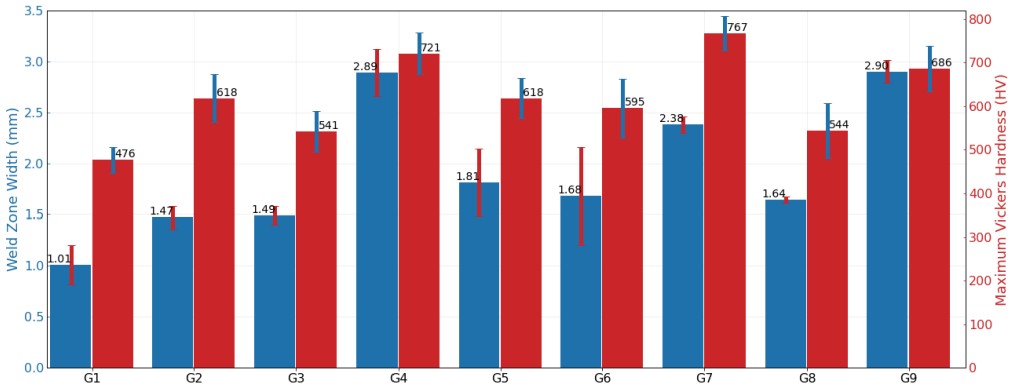

**Figure 5.** Weld zone width and maximum Vickers hardness of each welded joint.

On one hand, increasing the value of Fp enhances the axial pressure of the cylindrical rods, which is conducive to extruding the thermoplastic metal to form flash, thus diminishing the WZW. On the other hand, increasing the value of Fp increases the heat input at the being joined interfaces, which will enlarge the range of the high-temperature zone, thus leading to a broader WZW. It is considered that the latter has a more significant effect on the WZW. The WZWs of Groups 3 and 9 with larger set values of Fp (125 MPa) were broader than those of Groups 2 and 7 with smaller set values of Fp in the 3-Group with the same set values of Ss, respectively. On the premise of sound metallurgical bonding, welded joints with a narrower WZW are expected. As for the having been adopted process parameters, the low set value of Ss (1800 rpm), small set value of Fp (75 MPa), and large set value of Up (175 MPa) are recommended as the CDFW process parameters for the favorable WZW, that is, Group 1.

The detailed optical micrographs of a representative welded joint (specimen of rail head in Group 3) are given in Figure 6, where the microstructure corresponding to its different subzones are displayed. From the adjacent area of the BM to the center of the

weld zone, three different subzones could be distinguished, that is (i) the heat affected zone (HAZ); (ii) the thermo-mechanically affected zone (TMAZ); and (iii) the central weld zone (CWZ). The boundary of the weld zone and the BM is a hyperbolic shape due to the linear velocity variation along the radial direction of the cylindrical rods. The linear velocity of the outer region of the cylindrical rod was higher, and more frictional heat was generated, thus bringing a broader weld zone, whereas the linear velocity at the inner region approached zero gradually, and less frictional heat was generated, thus bringing a narrower weld zone; furthermore, the linear velocity at the center region was zero, no friction heat was generated, and all the heat was conducted from its surrounding area, thus producing the narrowest weld zone. Furthermore, the extrusion of the thermoplastic metal from the inner region to the outer region of the cylindrical rods also thickened the WZW of the outer region.

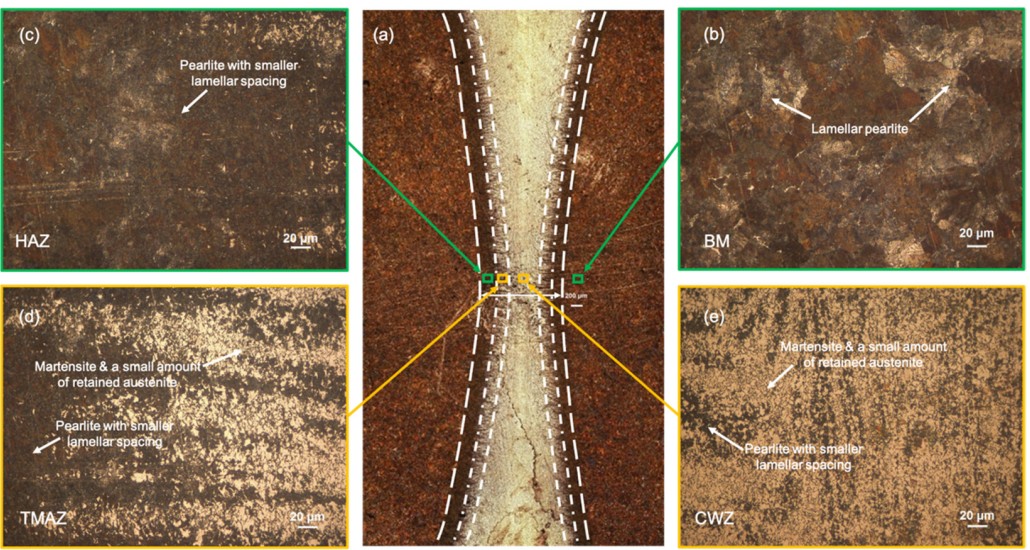

**Figure 6.** Optical micrographs of different subzones of CDFW joint. (**a**) Macrographic section, (**b**) BM, (**c**) HAZ, (**d**) TMAZ, (**e**) CWZ.

The microstructure of BM in Figure 6b exhibited a lamellar pearlite. Figure 6c shows the microstructure of HAZ, and the effect of the CDFW process is clearly visible: The welding heat failed to make this subzone reach the eutectoid transformation temperature. Thus, the microstructure was still pearlite but with reduced lamellar spacing due to the thermal process that goes through. The microstructure of CWZ in Figure 6e was made up of martensite, a small amount of residual austenite was formed through continuous dynamic recrystallization, and a part was pearlite with reduced lamellar spacing. This phenomenon resulted from the friction heat and the severe plastic deformation caused by the CDFW process as well as the high air-cooling rate. In TMAZ adjacent to CWZ, martensite and residual austenite content decreased, while the pearlite with reduced lamellar spacing increased, as illustrated in Figure 6d.

The EBSD phase maps and kernel average misorientation (KAM) maps corresponding to its different subzones of the same specimen as Figure 6 are shown in Figures 7 and 8, respectively. It could be observed from Figure 7 that the distribution of body-centered cubic (BCC) structure (ferrite and martensite phases) [34] with face-centered cubic (FCC) structure (residual austenite phase) across the interface had a significant change. In the subzones of BM and HAZ, almost no FCC structure was detected, and ferrite's BCC structures accounted for about 99.9%, as shown in Figure 7a,b. However, apparent structural phase transition occurred both in the subzones of TMAZ and CWZ. The proportion of FCC structures of residual austenite increased from 10% to 19%, respectively, as shown in Figure 7c,d.

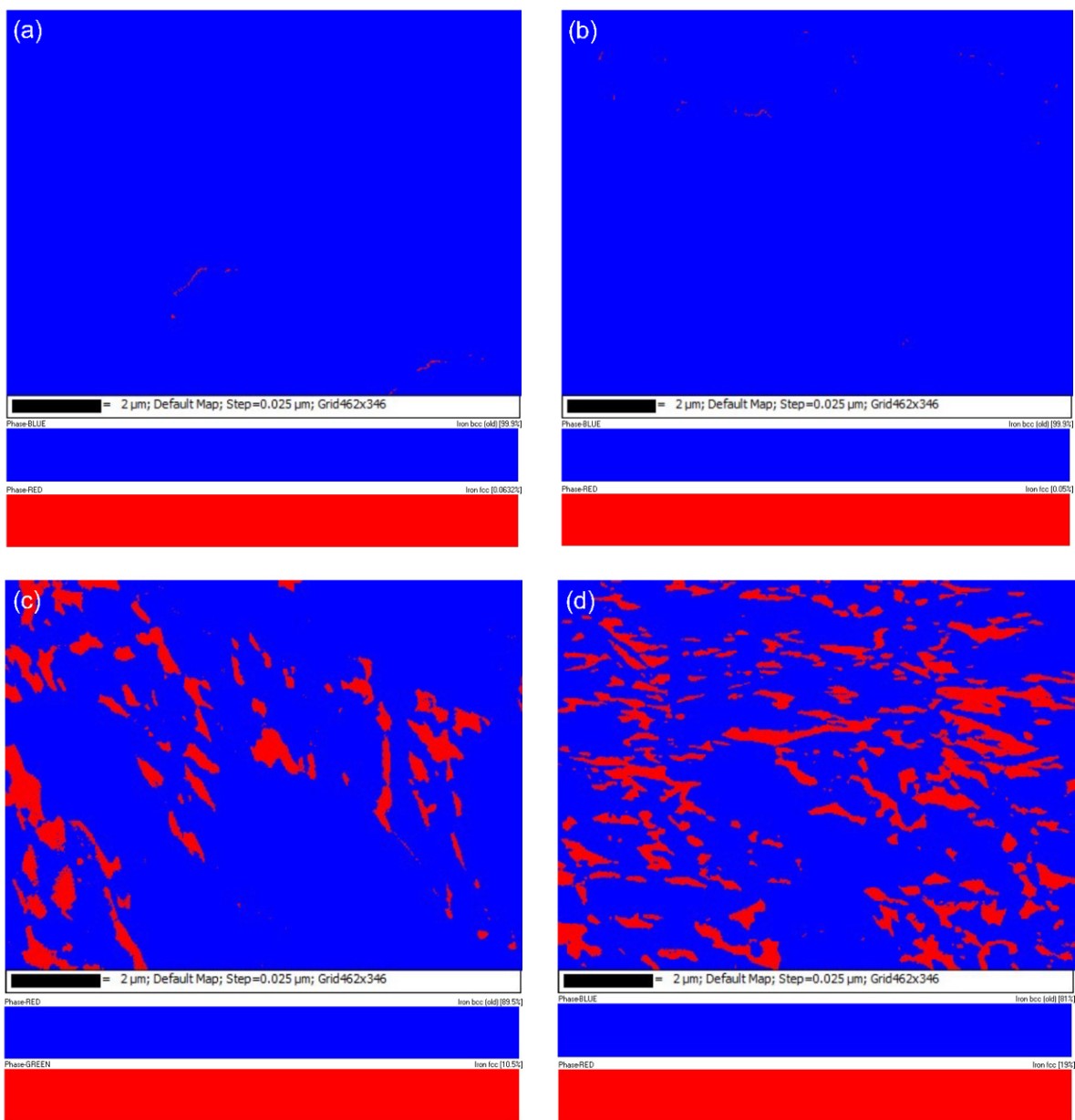

**Figure 7.** EBSD phase maps of Group 3. (**a**) BM, (**b**) HAZ, (**c**) TMAZ, (**d**) CWZ.

As martensite structures generally hold high internal stress and dislocation density [35], it is difficult to recognize through the phase maps, as shown in Figure 7. In contrast, the KAM maps in Figure 8 were used to distinguish the high degree of martensitic transformation by high KAM value from the corresponding BCC areas in their phase maps [36,37]. In the KAM maps of BM and HAZ, as shown in Figure 8a,b, respectively, hardly any high KAM value areas were observed, and most of the areas held low KAM value (blue areas), indicating that these subzones of BM and HAZ were not martensitized. However, the high KAM value areas were widely found in the subzones of both TMAZ and CWZ, as shown in Figure 8c,d, respectively, and more high KAM value areas were observed in CWZ (Figure 8d), indicating that martensite appeared in the subzones of both TMAZ and CWZ, except for ferrite and residual austenite. Generally speaking, the results of the EBSD analyses were consistent with the metallographic observation.

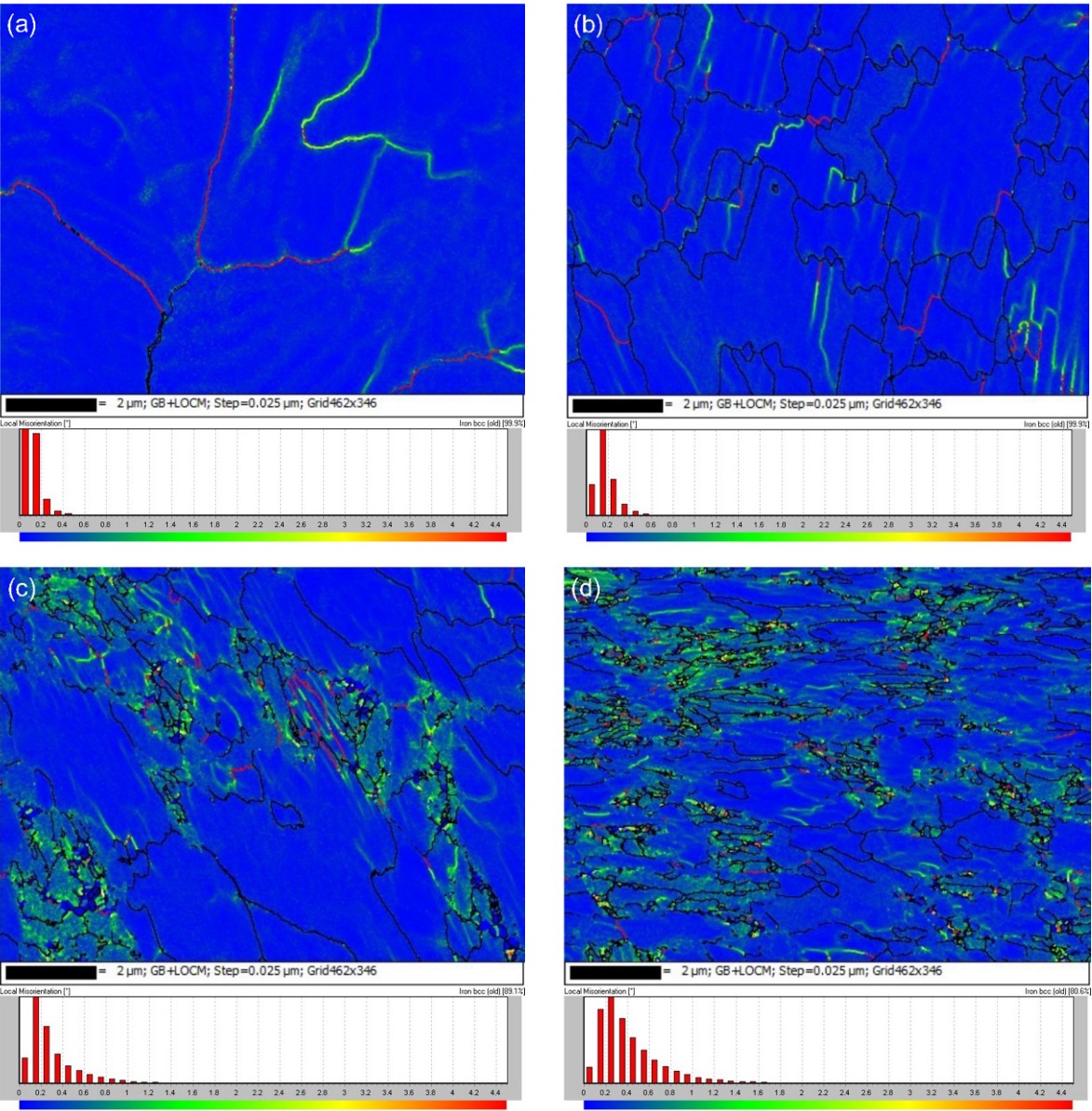

**Figure 8.** EBSD kernel average misorientation maps of Group 3. (**a**) BM, (**b**) HAZ, (**c**) TMAZ, (**d**) CWZ.

### 3.3. Hardness Measurement

Figure 9 shows the microhardness distributions of specimens at the welded joint center region for Groups 1, 4, and 9. It was observed that higher hardness zones occurred at the joined interface of all specimens, and their profiles exhibited similar tendencies. The maximum hardness level was recorded in the subzone of CWZ, and the hardness of the subzones of TMAZ and HAZ gradually decreased to that of BM. Several physical factors caused this increase in hardness: (i) martensite and residual austenite microstructures produced by continuous dynamic recrystallization during CDFW process, as discussed in Section 3.2; (ii) the grain refinement due to severe plastic deformation; and (iii) the presence of pearlite with reduced ferrite lamellar spacing, as described above (Section 3.2). The hardness of TMAZ was lower than that of CWZ, which corresponded to the content decrease in martensite and residual austenite microstructures. The hardness level of HAZ was still higher than that of the BM results from its fine grain microstructure induced by the thermal effect during the CDFW process [38]. It is worthwhile noting that the softening

zone, which generally appears in the FBW joints [5] and is prone to leading to low collapse defects, was not detected in the CDFW joints.

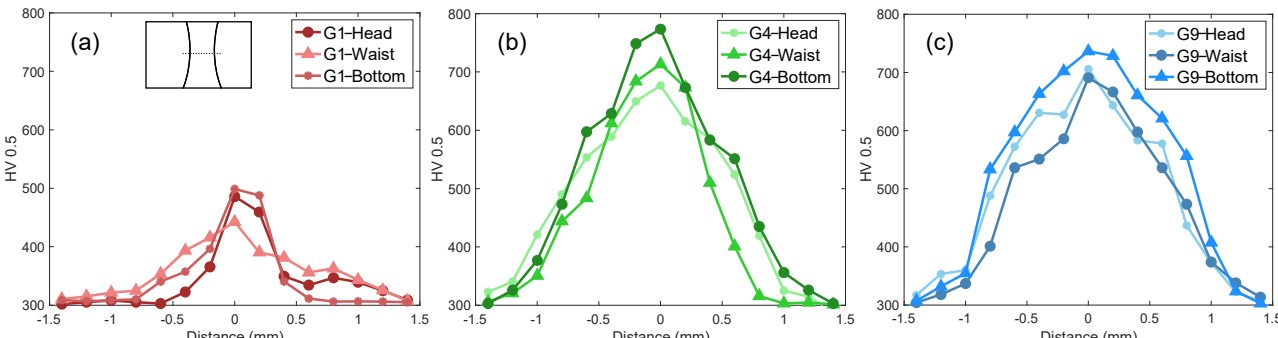

**Figure 9.** Vickers hardness curves along the direction perpendicular to the weld joining interface at the center area of specimens in Group 1 (**a**), Group 4 (**b**), and Group 9 (**c**).

The maximum Vickers hardness (MVH) values of specimens of the nine groups' welded joint are illustrated in Figure 5. The MVH values varied from $476 \pm 30$ HV$_{0.5}$ to $767 \pm 40$ HV$_{0.5}$. The effect of the set value of Ss on the MVH value was evident: the values of MVH in Groups 1–3 with a smaller set value of Ss became smaller than those of other groups with higher set values of Ss. As a matter of fact, the increase in the set values of Ss will consequently raise the temperature within the joined interface regions, generating a more thermoplastic metal, which inclines to form martensite microstructures at the high air-cooling rate, thus making the values of MVH increase, while the set value of Up generates the opposite influence. Groups 1, 6, and 8 with the maximum set value of Up (175 MPa) received the smallest values of MVH among the 3-Group with the same set values of Ss. In fact, the extrusion of the high-temperature thermoplastic metals by the high set value of Up resulted in a remarkable decrease in the value of MVH, which is favored by the CDFW joint properties.

On the premise of sound metallurgical bonding, the welded joints with the value of MVH close to that of the BM are favorable. In other words, as low as the set value of Ss decreases and as high as the set value of Up increases, the hardness values of all subzones of the welded joint reduce. As for the having been adopted process parameters, the low set value of Ss (1800 rpm), the large set value of Up (175 MPa), and the small set value of Fp (75 MPa) were the ideal process parameters for the performance of MVH, that is, Group 1, where the value of MVH (476 HV$_{0.5}$) was only 146.5% of that of the BM (325 HV).

### 3.4. Tensile Test

Figure 10 shows the representative stress–strain curves of specimens of the welded joints from the experiment of Group 8. No prominent necking stage could be observed in these curves, as the CDFW process made the original plastic material (U75V rail steel) have characteristics similar to a brittle material. Due to the discrepant mechanical properties for different parts of the steel rail (caused by the hot rolling process), the average tensile strength ($R_m$) and percentage elongation after fracture ($A$) of the welded joint specimen from rail waist were significantly lower than those of the welded joint specimens from the rail head and bottom. Therefore, the averaged values of the tensile test results of multiple specimens sampled from different parts (head, waist, and bottom) of steel rail are more comprehensive and valuable.

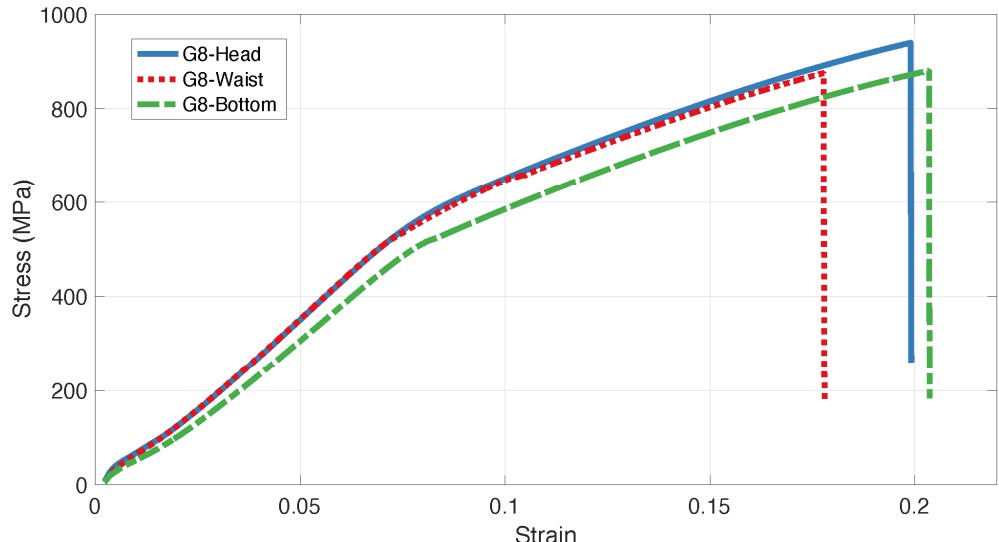

**Figure 10.** The strain–stress curves of Group 8.

The averaged values of $R_m$ and $A$ of all specimens are listed in Table 3. The fracture of all specimens occurred in the subzone of CWZ. Compared with the tensile properties of the BM ($R_m$ = 1062 MPa and $A$ = 29%), the values of $R_m$ and $A$ of the as-welded joint specimens were both decreased. However, these specimens had basically reached the demand value of $R_m \geq 880$ MPa for the U75V rail steel FBW joint specified in TB. Although the values of $R_m$ in Groups 5 and 6 were less than 880 MPa, they still exceeded 870 MPa (only 1% less than the expected value of $R_m$ in TB). Furthermore, the values of $A$ in all groups were almost three times the expected values of $A$ specified in TB ($A \geq 6\%$). In addition, the post-weld heat treatment (PWHT) is usually needed for the FBW joints [39] to meet the requirement of the expected value of $R_m$ specified in TB, which further shows that CDFW is an efficient and high-quality technology for rail steel welding.

**Table 3.** The tensile properties ($R_m$ and $A$) and impact toughness ($KU_2$) of each welded specimen.

| Group | $R_m$/MPa | vs. TB * | $A$/% | vs. TB * | $KU_2$/J | vs. TB * |
|---|---|---|---|---|---|---|
| 1 | 1001 | 114% | 21 | 356% | 3.56 | 55% |
| 2 | 935 | 106% | 19 | 322% | 2.00 | 31% |
| 3 | 905 | 103% | 20 | 328% | 2.39 | 37% |
| 4 | 937 | 106% | 21 | 344% | 2.29 | 35% |
| 5 | 870 | 99% | 18 | 300% | 1.91 | 29% |
| 6 | 871 | 99% | 17 | 289% | 2.57 | 40% |
| 7 | 884 | 100% | 18 | 300% | 2.11 | 32% |
| 8 | 899 | 102% | 19 | 322% | 3.08 | 47% |
| 9 | 908 | 103% | 18 | 300% | 3.37 | 52% |

* The ratio of $R_m/A/KU_2$ to TB is calculated and presented in the form of "vs. TB".

It could be considered that a low set value of Ss is expected. The values of $R_m$ obtained in Groups 1–3 with a small set value of Ss (1800 rpm) were higher than those obtained in other groups with high set values of Ss (2000/2200 rpm). The tensile properties of all specimens did not obviously vary with the axial pressure. The experiments in Group 1 with the low set value of Ss (1800 rpm), small set value of Fp (75 MPa), and large set value of Up (175 MPa) obtained the best tensile properties (1001 MPa, 94.3% of the BM), which suggests that the set values of process parameters in Group 1 were the ideal choice.

For the purpose of analyzing the fracture mechanism of as-welded joints, a typical specimen (G3) was selected for fracture investigation. The fracture morphologies are shown in Figure 11. The river patterns could be observed in the fracture initiation zone, and tear

edges without dimples are presented in the fracture propagation zone, indicating that the fracture morphology is a typical quasi cleavage fracture.

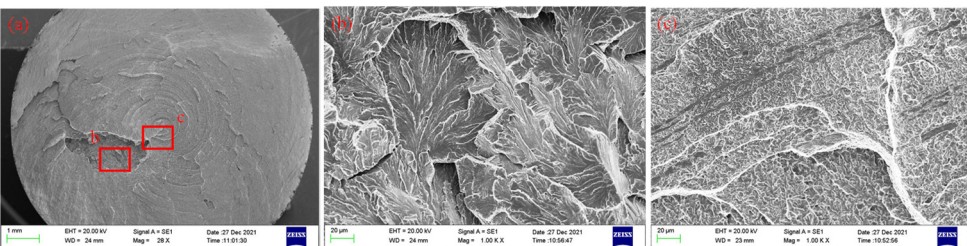

**Figure 11.** SEM fractography of a tensile specimen (G3): (**a**) overall morphology, (**b**) fracture initiation zone, (**c**) fracture propagation zone, (**b**,**c**) are the high magnification of (**b**,**c**) in (**a**).

### 3.5. Impact Toughness Test

The averaged values of the impact absorption energy ($KU_2$) of the nine group experiments are listed in Table 3, where $KU_2 = KU_{2(5.0)}/0.5$ [40]. The impact fracture of all specimens appeared in the subzone of CWZ, where the u-notch of the small size impact specimen was located. Compared with the $KU_2$ value of U75V rail steel (12.49 J), the values of $KU_2$ of all as-welded joint specimens were significantly decreased. Unsatisfactorily, their $KU_2$ values failed to meet the expected $KU_2$ value for the U75V rail steel FBW joint specified in TB ($KU_2 \geq 6.5$ J). The reasons are that the brittle and hard martensite microstructures that appeared in the subzones of CWZ and TMAZ make them easy to break and fractured by an external force, and the residual stress produced by the thermo-mechanical coupling effect during the CDFW process.

It is noticeable that the increase in the set value of Up is conducive to the extrusion of brittle and hard martensite microstructures in the weld zone, so the $KU_2$ values of the groups with the large set values of Up were relatively high. Moreover, under the condition of sound metallurgical bonding, reducing the set values of Ss and Fp were able to decrease the welding heat input, thus cutting down the generation of thermoplastic metal. The $KU_2$ value of Group 1 (3.56 J) was still the highest among all groups.

The fracture mechanism of the impact specimen was also studied. The same specimen (G3) was used for observation, as depicted in Figure 12. Similar to the fracture mechanism of the tensile specimen, river patterns could also be observed in the fracture initiation zone of the impact specimen, and the fracture propagation zone presented tear edges. Still, no dimples were observed, indicating the quasi-cleavage fracture.

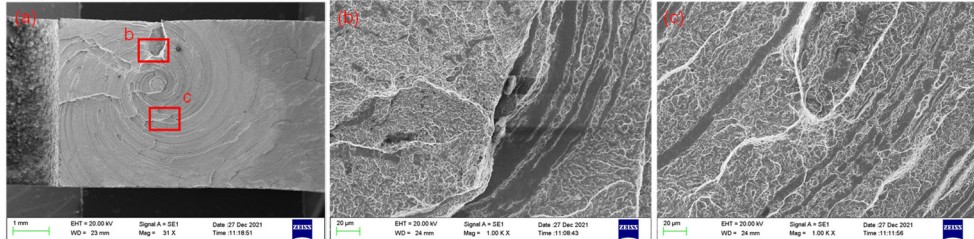

**Figure 12.** SEM fractography of an impact specimen (G3): (**a**) overall morphology, (**b**) fracture initiation zone, (**c**) fracture propagation zone, (**b**,**c**) are the high magnification of (**b**,**c**) in (**a**).

## 4. Discussion

The above-exhibited results convincingly demonstrate that the CDFW process parameters significantly influenced the microstructure and mechanical properties of the CDFW joints. The set values of Ss and Fp determine the welding heat input, which governs the generation of thermoplastic metal and the eutectoid transformation while the axial pressure including Fp and Up, influences the grain refinement and the extrusion of thermoplastic

metal, and thus affect the flash formation. The complex thermo-mechanical coupling effect produced by process parameters leads to microstructural transformation in different subzones of the weld zone and, furthermore, gives rise to the variation in the mechanical properties within the welded joints.

Low set values of Ss (1800 rpm) and Fp (75 MPa) reduced the welding heat input, thus cutting down the production of thermoplastic metal; the large set value of Up (175 MPa) extruded more thermoplastic metal into flash and made the grains finer within the weld zone. Under this optimal process parameters group, the value of $R_m$ was the highest (1001 MPa, 94.3% of the BM), the value of $A$ was the largest (21%, 3.5 times of that specified in the TB), the value of $KU_2$ was also the highest (3.56 J, 54.8% of that specified in the TB), the value of MVH was the lowest (476 $HV_{0.5}$, 146.5% of the BM), and the value of WZW was the narrowest (1.01 mm, only 2.5% of that of FBW joint) among all experimental groups.

It could be seen from the experimental results that the tensile strength of the CDFW joint considerably exceeds that of the LFW joint in the literature [28], which is only 74.8% of the BM. Although the impact toughness of the CDFW joint slightly surpassed that of the LFW joint, none of them met the requirements for FBW joints specified in TB ($KU_2 \geq 6.5$ J). These results are because some hard and brittle martensite microstructures appeared in the weld zone due to the local temperature exceeding the phase-transition temperature and rapid cooling rate. In addition, the fracture analysis of tensile and impact specimens showed that the fracture mechanism was a quasi-cleavage fracture, further confirming the existence of these brittle microstructures. Therefore, one of the important research directions of the welding of U75V steel rail by CDFW in the future is to reduce the maximum temperature within the weld zone during the friction process by process parameters optimization (e.g., decreasing the values of Ss or Fp) to further enhance the impact toughness of welded joints.

## 5. Conclusions

In this article, the cylindrical rods with a diameter of 12 mm sampling from U75V steel rail were welded by the CDFW method, and the influence of the process parameters such as spindle speed, friction pressure, and upsetting pressure on the mechanical properties as well as the microstructure of the as-welded joints was investigated. The main conclusions of this research are summarized as follows:

1.  CDFW is an effective method to realize reliable welding of U75V rail steel, as sound metallurgical bonding without welding defects (e.g., voids, cracks, or slag inclusions) is formed in each weldment. The welding process only takes a few seconds, and the axial shortening length is only about 5 mm. These are much shorter than those in the FBW method (with a welding period of more than 1 min and an axial burning length of more than dozens of millimeters).
2.  Microscopic observation and EBSD analyses show that BM comprises lamellar pearlite; the HAZ consists of lamellar pearlite with reduced ferrite lamellar spacing; martensite and residual austenite microstructures appeared in the TMAZ and CWZ. From TMAZ to CWZ, the content of martensite and residual austenite microstructures increased while the content of pearlite with reduced ferrite lamellar spacing microstructures decreased.
3.  Process parameters determine the microstructure and mechanical properties of the as-welded joints by affecting heat generation, maximum temperature, metal plasticizing, and flash extrusion. The small set values of Ss (1800 rpm) and Fp (75 MPa) as well as the large set value of Up (175 MPa) were the preferred process parameters group (Group 1). In this group, the value of $R_m$ was 94.3% of that of the BM, and the value of $A$ was 3.5 times that specified in the TB. Furthermore, the values of $KU_2$ (3.56 J), MVH (476 $HV_{0.5}$), and WZW (1.01 mm) were the largest, lowest, and narrowest ones among all experimental groups, respectively.
4.  The tensile properties of the CDFW joint met the corresponding demands for that of the FBW joint specified in TB. However, the value of impact toughness decreased significantly compared with that of the BM, nor failed to meet the expected value of

$KU_2$ specified in TB (54.8% of the demanded value) for the FBW joint. The fracture mechanism of the tensile and impact specimens was quasi-cleavage fracture. The presence of the brittle and hard martensite microstructures led to these phenomena. These microstructures also increase the microhardness of the weld zone, especially the joining interface, while it is satisfactory that the softening zone generally appeared in the FBW joints was not detected.

**Author Contributions:** Conceptualization and methodology, H.Z. and Z.Z.; Investigation, H.Z., C.L. and Z.Z.; Formal analysis, H.Z.; Writing—original draft preparation, H.Z.; Writing—review and editing, H.Z., C.L. and Z.Z.; Supervision, Z.Z. All authors have read and agreed to the published version of the manuscript.

**Funding:** This research was funded by the National Natural Science Foundation of China under grant nos. 51775301 and 51075231.

**Institutional Review Board Statement:** Not applicable.

**Informed Consent Statement:** Not applicable.

**Data Availability Statement:** Not applicable.

**Acknowledgments:** We would like to thank Tongyi Li and Yuan Gao for their help in the welding experiments and thank Jianguo Li and Xin Yang for their help in the EBSD and fracture analysis.

**Conflicts of Interest:** The authors declare no conflict of interest.

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
