# Peer review of "Influence of CDFW Process Parameters on Microstructure and Mechanical Properties of U75V Rail Steel Welded Joint"

_metals, doi:10.3390/met12050711_

Round 1

Reviewer 1 Report

Dear authors, the article is of good quality but I have some important remarks to refer to.
The authors stated in the introduction that the CDFW method is designed for rail splicing. In the article the authors use round bars with a diameter of 12mm. The rails are much larger and have an irregular shape that must be rotated at a speed of 2000 rpm with appropriate pressure. I do not think it is a good solution to connect the rails together using the CDFW method. Please provide a literature reference that includes an example of joining large-sized elements using the CDFW method.

The authors have well described the microstructure of the connection and determined the tensile strength of the materials. However, scanning images of the specimen fractures are missing. Please also make a run of the failure of the specimens due to monotonic stretching.

Kind regards

Author Response

Point 1: The authors stated in the introduction that the CDFW method is designed for rail splicing. In the article the authors use round bars with a diameter of 12mm. The rails are much larger and have an irregular shape that must be rotated at a speed of 2000 rpm with appropriate pressure. I do not think it is a good solution to connect the rails together using the CDFW method. Please provide a literature reference that includes an example of joining large-sized elements using the CDFW method.

Response 1: Thank you very much for kindly reviewing the manuscript we submitted and offering us many great constructive and helpful suggestions. We have fully considered these suggestions and carefully revised the original manuscript according to these suggestions.

For this comment, we give the answer from the following two aspects. Firstly, we cite new references [28] and [30] as examples of joining large-sized elements using some type of FW methods in Section 1 to support that the CDFW method could potentially be applied in the joining of rails with larger section sizes. Literature [28] shows the CDFW of D50Re steel workpieces with a diameter of 100 mm (Line 89 in Section 1). Literature [30] realizes linear friction welding of the rail steel workpiece with the interface size near the 60 kg/m type rail with a height of 176 mm (Line 104 in Section 1).

Secondly, just as the comment from the reviewer, it is indeed difficult or even impossible to adopt the traditional CDFW method to join the rails with a section of much larger size and an irregular shape under the rotation speed of 2000 rpm with appropriate pressure. The ultimate goal of our research work is to develop a new and innovative kind of friction welding process method and applicable equipment for steel rail welding to obtain better welded joint performance. This innovative equipment for steel rail welding has applied for a patent for an invention in China. According to preliminary research and calculation, the rotating speed of the workpiece for steel rail CDFW is less than 1000 rpm. This paper presents only the preliminary exploration for applying the CDFW technology to the joining of U75V rail steel and further investigates the influence of CDFW process parameters on the microstructure and mechanical properties of the welded joint, expecting laying the foundation for the CDFW process parameters optimization and equipment detailed design of successful joining of steel rail.

Point 2: Scanning images of the specimen fractures are missing. Please also make a run of the failure of the specimens due to monotonic stretching.

Response 2: Thanks for this valuable suggestion. After further tests, we add the fracture analysis for tensile (Line 442 in Section 3) and impact (Line 472 in Section 3) specimens in G3.

Thank you very much again for all your valuable time and help!

Reviewer 2 Report

Dear Authors,

first of all, I would like to thank you for submitting the manuscript to the Metals journal.

The manuscript deals with the research of the weldability of the U75V rail steel with continuous-drive friction welding (CDFW) process. The influence of chosen welding parameters (spindle speed, friction and upsetting pressure) on the presence of defects, macro and microstructure, phase evaluation, tensile strength and impact toughness of weldments was evaluated. Authors found that the more favourable results could be obtained at lower values of spindle speed and friction pressure that cause less heat input and plastic deformation. The topic of the manuscript seems to be very interesting and industrially very important because it has a direct practical application in the railway sector and could improve the economy of rails joining. 

The introduction provides sufficient theoretical background. All references used in the manuscript are relevant. Methods used during the experimental work are described adequately and in detail. The structure of the experiment is defined in logical sequence. Various and quite high number of experimental techniques were used for analysis of the welded samples. The results are clearly presented. However, I miss a better comparison with findings of other authors. I also recommend to use a scanning electron microscope to distinguish the structural features (f.e. lamellar pearlite) at higher magnifications, and thus resolution, in future work.

Based on above, I have only few additional questions, or suggestions for improvements. So, I recommend to publish the article after the minor revision.

Line 122 - In the section „2. Material and experimental methods“ I recommend to add a scheme of the welding for better description. It would be also good to mark in the scheme the spindle speed, friction and upsetting pressure (with arrows), if it is possible.

Line 122 - In the section „2. Material and experimental methods“ I recommend to more specify the used experimental equipment. I missed the the type, producer name, city and country in the case of the CDFW machine, hardness tester, light microscope, etc. It has to be improved in according to the Metals journal.

Line 267 - Why authors chose the sample G3 for EBSD observations.

Best regards

Author Response

Point 1: In the section 2. Material and experimental methods. I recommend to add a scheme of the welding for better description. It would be also good to mark in the scheme the spindle speed, friction and upsetting pressure (with arrows), if it is possible.

 Response 1: Thanks very much for your kindly helpful review of our manuscript. A scheme of the CDFW process has been added (Fig.1) for a better description. The spindle speed, friction and upsetting pressure have also been marked (Line 168 in Section 2).

Point 2: In the section 2. Material and experimental methods. I recommend to more specify the used experimental equipment. I missed the type, producer name, city and country in the case of the CDFW machine, hardness tester, light microscope, etc. It has to be improved in according to the Metals journal.

Response 2: Thank you for your detailed suggestion. We have supplemented the types, producer names, cities, and countries of the CDFW machine (Line 138 in Section 2), microhardness tester (Line 184 in Section 2), optical microscope (Line 189 in Section 2), EBSD system (Line 191 in Section 2), SEM for EBSD analysis (Line 192 in Section 2), tensile tester (Line 196 in Section 2), impact testing machine (Line 208 in Section 2), scanning electron microscope (Line 192 in Section 2), and SEM for fracture analysis (Line 216 in Section 2).

Point 3: Why authors chose the sample G3 for EBSD observations.

Response 3: Thank you for your careful review. Because G3 is a low set value of Ss group, and it is not the favorable one. It is convenient for us to study the overall situation of all groups (martensitic transformation in the center weld zone), as G3 is more representative.

Thank you very much again for all your great valuable time and help for us!

Round 2

Reviewer 1 Report

Dear authors, I accept your clarifications, and the corrections made.
However, please inform the reader in the introduction or preface about the final purpose of conducting the research. I understand that you have a series of studies to conduct on the basis of which you will create an innovative joining technology. You should write about it.

Author Response

Point 1: Please inform the reader in the introduction or preface about the final purpose of conducting the research. I understand that you have a series of studies to conduct on the basis of which you will create an innovative joining technology. You should write about it.

Response 1: Thank you so much for offering us many great constructive suggestions. We have seriously considered your recommendations and have made revisions to our manuscript.

The final purpose of conducting the research has been added in Line 119, Section 1.

Thank you very much again for all your great valuable time and help for us!
